

# Sedapp: a non-linear diffusion-based forward stratigraphic model for shallow marine environments

Jingzhe Li[1, 2], Piyang Liu[3], Shuyu Sun[4] (✉), Zhifeng Sun [2, 1], Yongzhang Zhou[5], Liang Gong[6], Jinliang Zhang[7], Dongxing Du[1,2]

1. College of Electromechanics, Qingdao University of Science and Technology, Qingdao 266061, China

2. Geo-Energy Research Institute, Qingdao University of Science and Technology, Qingdao 266061, China

3. School of Science, Qingdao University of Technology, Qingdao, 266520, China

4. King Abdullah University of Science and Technology, Jeddah 23955-6900, Saudi Arabia

5. School of Earth Science and Geological Engineering, Sun Yat-sen University, Guangzhou 510275, China

6. School of New Energies, China University of Petroleum (EastChina), Qingdao 266555, China

7. Department of Geography, Beijing Normal University, Beijing 100875, China

Correspondence to:   Shuyu Sun (frank.sun.sa@gmail.com)

## Abstract

The formation of stratigraphy in shallow marine environments has long been an important topic within the geologic community. Although many advances have been made in the field of forward stratigraphic modelling (FSM), there are still some shortcomings to the existing models. In this work, the authors present our recent development and application of Sedapp: a new non-linear open-source R code for FSM. This code uses an integrated depth-distance related function as the expression of the

transport coefficient to underpin the FSM with more along-shore details. In addition to conventional parameters, a negative-feedback sediment supply rate and a differentiated deposition-erosion ratio were also introduced. All parameters were implemented in a non-linear manner. Sedapp is a 3D (2DH) tool while also capable of 2D (1DH) scenarios. Two simplified case studies were conducted. The results show that Sedapp can not only assist in geologic interpretation, but is also an efficient tool for internal

architecture predictions.

**Keywords**: Forward stratigraphic modelling, continental shelf, R codes, fluvial-deltaic, continental fault basin



## 1 Introduction

Shallow marine areas are among the most active environments for sedimentation, where sea level,

tectonism, climate all influence the interactions between land and sea. Stratigraphic formation in this

environment has long been an important topic within the geoscience community, which has directly

resulted in the emergence of sequence stratigraphy (Haq et al., 1987; Li et al., 2015; Catuneanu, 2019).

Traditional qualitative methods on this subject have made great advances in the past half century, but it

is difficult to test the validity and internal consistency of a new concept and are less likely to raise

counter-intuitive ideas which may actually be true (Burgess, 2012; Burgess and Prince, 2015).

Computer modelling can help resolve this problem. The methodology is usually called forward

stratigraphic modelling (FSM) (Griffiths and Hadler-Jacobsen, 1995; Dalman and Weltje, 2012;

Sangster et al., 2019), although it was also called stratigraphic forward modelling (Burgess, 2006;

Sacchi et al., 2015; 2016; Ding et al., 2019), basin filling modelling (Syvitski and Hutton, 2001; Hutton

and Syvitski, 2008; Li et al., 2020), and stratigraphic simulation (Rivenaes, 1992; 1997; Lawrence et al.,

1990), etc.

FSM deals mainly with long-term geomorphologic/stratigraphic dynamics (Paola, 2000). It is

slightly different from sediment fluid-flow models, which deal more with the fluid-flow dynamics by

solving modified Navier-Stokes equations within a full study domain (generally shallow water

equations, e.g. HydroSedFoam of Zhu et al., 2019 or Delft3D in Ramos et al., 2019). FSM models can

be classified into two types, i.e. rule-based models (based on geometric or fuzzy logic) and

equation-based models (Paola, 2000; Syvitski and Hutton, 2001; Burgess et al., 2012; Sacchi et al.,

2015). The first type easily captures essential features and is less time-intensive, while it does a

relatively poor job of demonstrating predictability and revealing the physical processes (Strobel et al.,

1989; Kendall et al., 1991; Burgess, 2012). The latter type is also known as deductive models, which

are process-based and solve governing equations (Kaufman et al., 1991; Rivenaes, 1992; Granjeon and

Joseph, 1999; Griffiiths et al., 2001; Hutton and Syvitski, 2008; Li et al., 2020). For these long-term

processes, sediment flux is usually assumed to be proportional to the topographic gradient. Thus, a

diffusion equation like Eq. (1) is generally used to derive the governing equations in FSM models

(Salles et al., 2018; Ding et al., 2019).





$$\frac{\partial h}{\partial t} = \nabla \cdot (\Gamma \nabla h)$$

(1)

where $h$ denotes the topography, $t$ denotes the time and $\Gamma$ denotes the transport coefficient.

Diffusion-based models are good at modeling scaled stratigraphic sequence (relative larger scales, e.g. clinoform formation) processes. $\Gamma$ in Eq. (1) can be defined using different values for different environments (Zhang et al., 2020). Various $\Gamma$ types are used based on different needs and environments (Rivenaes, 1997; Zhang et al., 2020). Models with constant $\Gamma$ values are usually called linear models; otherwise, they are known as non-linear models.

Although the sediment diffusion assumption is considered a practical representation of long-term slope processes, it is still too simplistic when the $\Gamma$ is used as a constant because natural agents such as air and water, phenomena such as mass wasting, and biological agents actually move sediment at rates that are not determined solely by slope (Salles et al., 2018). This severely limits the application scope of linear diffusion-based models. On the contrary, non-linear models are relatively more flexible. Many non-linear models define the transport coefficient using water depth-related functions (e.g., Clarke et al .1983; Kaufman,1991). These water depth models work well in general coastal zones. However, in the shallow marine environments with river injection, the water depth models are usually not applicable. Depositional processes around the river mouth are more active than those at a distance, even when they are at the same water depth. Hence, it is difficult for water depth models to reveal along-shore variability, especially in 3D scenarios (actually 2D-H, with two horizontal dimensions and the elevation H). Because of this reason, many hybrid hydrodynamic-diffusion models were proposed. For example, river plume was introduced to differentiate the suspended sediment fluxes along the coastline (Syvitski and Hutton,2001; Hutton and Syvitski, 2008; Dalman and Weltje., 2012). This treatment realized the modelling of the convex shapes of delta out from the river mouths. However, the computational load was also significantly increased because of the newly added advection-diffusion process. The introduction of many hydrodynamic parameters also increased the difficulty of its usage.

In addition, many existing models are not free or open-source, making it difficult for people to reproduce and improve them. Some codes, although free, can only be used in Linux systems, which makes them inconvenient to use in most PC terminals.

In this paper, we propose a new non-linear model, which is expected to overcome the shortcomings of the existing models. Along with some other features, this model is integrated into a



framework called Sedapp, which is an open-source and cross-platform application written in R. We use

examples to show how this model works and test its effectiveness and convenience in reconstruction of

sedimentary systems, revealing their internal architectures.

## 2 Methodology

### 2.1 Mathematical model

The Sedapp mathematical model can be expressed as follows:

$$F_i \frac{\partial h}{\partial t} = \max\left( \nabla \cdot (\Gamma_i \nabla h), \frac{1}{Der} \nabla \cdot (\Gamma_i \nabla h) \right) + q \tag{2}$$

$$\sum_{i}^{n} F_i = 1 \tag{3}$$

where $F_i$ is the fraction of the $i$th class of lithology, h is elevation, $t$ is time, $\nabla$ is the nabla operator,

$Der$ is a user-defined parameter denoting the ratio of deposition to erosion (it can be a scalar, vector or

tensor value depending on its temporal and spatial variability), $\Gamma_i$ is the diffusion coefficient for the $i$th

class of lithology, and q is the source term that is a function of coordinates and time (the source term is

used only for endogenetic sedimentation, especially carbonates. If endogenetic sedimentation is

ignored, the source term can be left out). Among them, $h$ and $F_i$ are the primary unknowns.

Note that $\Gamma_i$ cannot be outside the parentheses, because they are not constants but rather functions

of spatial coordinates and time. The expression of a general $\Gamma$ can be expressed as:

$$\Gamma = \max(\alpha \, e^{-\frac{D(\mathbf{x},t)^\eta}{\beta}}, \quad \alpha_{wd} e^{-\frac{Wd(\mathbf{x},t)^{\eta_{wd}}}{\beta_{wd}}}) + \varepsilon \tag{4}$$

where $\alpha/\alpha_{wd}$ are preexponential factors ($L^2/T$) , $\eta/\eta_{wd}$ are distance indexes (no dimension), $\beta/\beta_{wd}$ are

spatial scale factors ( $L^\eta$  or  $L^{\eta_{wd}}$ ), and ε is an adjustment factor ($L^2/T$) reflecting the environment

energy. In particular, distance function $D=D(\mathbf{x},t)$ and water depth function $Wd(\mathbf{x},t)$ change with spatial

coordinates and time, and they work for the marine portion only.

When $Der = 1$ and n = 2, the 3D (actually 2DH, because h is another dimension perpendicular to

x and y) scenario for Eq. (2) and Eq. (3) can also be expressed as:

$$F \frac{\partial h}{\partial t} = \frac{\partial}{\partial x}\left( \Gamma_1 \frac{\partial h}{\partial x} \right) + \frac{\partial}{\partial y}\left( \Gamma_1 \frac{\partial h}{\partial y} \right) + q(x,y,t) \tag{5}$$



$$(1-F)\frac{\partial h}{\partial t} = \frac{\partial}{\partial x}\left(\Gamma_2 \frac{\partial h}{\partial x}\right) + \frac{\partial}{\partial y}\left(\Gamma_2 \frac{\partial h}{\partial y}\right) + q(x,y,t) \tag{6}$$

where x and y are spatial coordinates. This is especially suitable for cases dealing only with two classes

of lithology for simplicity, where $\Gamma_1$ is the transport coefficient for sand and $\Gamma_2$ is the transport

coefficient for mud.

For 2D (1D-H) scenarios, especially along the section line through the river mouth, the distance

related term is generally larger than the water depth related term, so the latter term within the max

function in Eq. (4) is usually omitted. For convenience in coding, also ignoring the endogenetic

sedimentation, Eq. (5), Eq. (6) and Eq. (4) can be simplified into:

$$F\frac{\partial h}{\partial t} = \frac{\partial}{\partial x}\left(\Gamma_1 \frac{\partial h}{\partial x}\right) \tag{7}$$

$$(1-F)\frac{\partial h}{\partial t} = \frac{\partial}{\partial x}\left(\Gamma_2 \frac{\partial h}{\partial x}\right) \tag{8}$$

$$\Gamma_i = \alpha_i \cdot e^{-\frac{(c \cdot D(\mathbf{x},t))^2}{E}} + \varepsilon, i = 1,2 \tag{9}$$

The joint effect of $c$ and $E$ in Eq. (9) is equivalent to that of $\beta$ in Eq. (4). The variable $c$ here, with

a dimension of $L^{-1}$, is mainly used to facilitate the scale of distance and differentiate the transport

characteristics of different sediment types (e.g., sand and mud). E is a dimensionless constant that

represents hydraulic characteristic energy.

**2.2 Code Implementation**

Sedapp was written in the R language and its solution procedure was based on the finite volume

method (FVM), which has the desired property of local mass conservation and has a clear physical

meaning (Versteeg and Malalasekera, 2007; Moukalled et al., 2016; Liu P. et al., 2017). The

cell-centered variable arrangement method was used to store the unknowns at the grid element

centroids. The non-linearity was implemented through stepwise iteration (Fig.1).

The brief work-flow within a single time step is as below:

1) Implement user-defined tectonic subsidence and update the topography;

2) Implement user-defined sea level and identify/update the shoreline location;

3) Solve the differential deposition/erosion function;

4) Implement the compaction and isostatic subsidence.

Step 3) is a major step. According to the hypothesis of diffusion-based FSM models, the change

rate (by either deposition or erosion) is proportional to the gradient of the slope (Fernandes et al., 1997;

Pelletier, 2013). If we use the diffusion equation/law directly without any differential treatments

between deposition and erosion (in other words, Der is held at 1), it will be contrary to the geological

knowledge that deposition and erosion processes are two distinct processes with different rates. Hence,

the max() function is used as in Eq. (2). Generally, the erosion process occurs at a different rate than

deposition (also called erosion constraints, see Galy and France-Lanord,2001), so Der is usually not

equal to 1. For example, if we wanted the erosion rate to be only 1/100 of the deposition rate, the Der

can be set to 100. In this case, if it is a deposition process (namely the $\frac{\partial h}{\partial t} > 0$), $\nabla \cdot \left( \Gamma_i \nabla h \right)$ would be

larger than $\frac{1}{der} \nabla \cdot \left( \Gamma_i \nabla h \right)$, and $\nabla \cdot \left( \Gamma_i \nabla h \right)$ is used. Otherwise, the $\frac{1}{der} \nabla \cdot \left( \Gamma_i \nabla h \right)$ is used. If a

non-erosion case is desired, Der can be set to a very large value.

Generally, sediment supply rate cannot be directly defined through boundary condition settings,

since the latter can only determine the boundary slope. Therefore, Sedapp uses a negative-feedback

strategy to define the sediment supply rate. At each time step, the total amount of deposition within a

step is first calculated using the previously defined $\alpha_{test}$, and then the adjusted $\alpha_{mod}$ is calculated by

Eq. (10):

$$\alpha_{mod} = \alpha_{test} \frac{V_{expected}}{V_{test}}$$    (10)

where $\alpha_{mod}$ denotes the modified α of this time step; $V_{expected}$ denotes the expected sediment increment,

namely the sediment supply rate; and $V_{test}$ denotes the computed sediment increment with $\alpha_{test}$.

## 3 Characteristics

### 3.1 Nonlinear transport coefficients

The nonlinear transport coefficient is a feature of Sedapp. Sedapp's transport coefficient uses a

function of both the distance from the estuary and the water depth. This feature makes it easier to

simulate fluvial-deltaic processes in 3D scenarios, which can reflect changes along the shore. Even in

2D cases, this feature also has some advantages (see the discussion section for details).

Generally, a smaller $c$ value results in higher sediment travel distance and a larger distribution

range when the total amount of sediment is fixed. For example, the c of mud is usually set to 50%-85% of sand, thus reflecting the differential deposition of sand and mud. In addition, the environment energy $\varepsilon$ can also influence the sediment travel distance that a larger $\varepsilon$ can make the sediment travel further. As sedimentation progresses, the position of the estuary may change, so the distance from the estuary is

updated at each time step to achieve the nonlinearity of $\Gamma$.

### 3.2 Differential and customizable deposition/erosion rate

During the actual deposition process, the properties of the lower strata (such as compaction degree, lithology, and age, etc.), as well as some external environmental factors (such as temperature, humidity and pH value, etc.), will affect the erosion rate. Therefore, the customized treatment of

erosion rate is another Sedapp characteristic.

In Sedapp, the deposition rate is a parameter that can be specified directly (for the adjustment process see section 2.2). Furthermore, the Der parameter is a user-defined parameter that controls the ratio of deposition rate to erosion rate. When Der is 1, the deposition rate is equal to the denudation rate (Fig.2a), and when Der value is 10 or 100, denudation is significantly weakened (Fig.2b). Theoretically,

if the value of Der is large enough, it is equivalent to completely eliminating the denudation effect. Der values should be customized according to the actual situation.

### 3.3 Customizable compaction

Compaction is an important geological process after sediment deposition, especially when the sediment thickness is very high. In Sedapp, the compaction process can be easily realized by setting the

composition of lithology and porosity curves.

In this paper, we designed a pyramid-shaped mountain simulation commonly used by other researchers (as shown in Fig.3, see Rivenaes,1992 and Yuan et al.,2019 for reference). The Der value was set to 1. The sediment supply ratio of sand and mud was set to 1:1, and the porosity curve was set as shown in Fig.3d. After simulation, the top of the pyramid was denuded and the foot of the pyramid

had deposited sediment of a given thickness.

To illustrate the effect of compaction, Sedapp introduces a scale factor that can enlarge the longitudinal scale. Fig.3a shows the original compression scale (that is, the scale factor was equal to 1), and the scale factors in Fig.3b and Fig.3c were 100 and 1000, respectively. It can be seen that sediment



thickness at the foot of the pyramid in Fig.3c was significantly smaller than that in Fig.3a. The factors

that caused these differences were not only depth but also the proportion of sand and mudstone and the

shape of depth-porosity curves, which can be easily adapted to different scenarios by modifying the

lithologic proportion and porosity-depth functions in Sedapp.

## 4 Verification of Sedapp

To identify how well the algorithm works within geological context, some simple benchmark

simulations are given below.

### 4.1 Typical stacking patterns

Typical stacking patterns including forced regression, normal regression, and transgression can

be formed (Fig.4) by fixing sediment supply while controlling the adjusted sea level rise rate.

During the period of sea-level decline, the shoreline moved seaward, and the onlap points also

moved seaward and form the offlap and downlap stratigraphic termination structures (Fig.4a). During

slow sea-level rise, the shoreline continued to move seaward, but the onlap points started to move

landward, forming an onlap termination structure. At the other end, the downlap structure continued to

exist. During rapid sea-level rise, the shoreline started to move landward and the onlap points also

moved landward. At this time, downlap structure did not exist above the slope break, but may have

existed below the slope break.

### 4.2 Typical two-cycle scenario

To demonstrate the complete base level changing process, this paper designed a simulation with

two full sinusoidal cycles as shown in Fig.5. In the first cycle, the shoreline dropped and moved

seaward. Then it slowly rose and gradually moved landward until it reached the highest point and

tended to stabilize. The water depth of deposition in the strata gradually deepened from left to right on

the marine side (Fig.5a), and the sandy content reached a maximum around the shoreline (Fig.5b) near

the shoreline. In the strata on the land side, the sand content was stratified. The sand content was

relatively large during the early transgression and subsequently relatively small. The second cycle was

located above the first cycle and continued the same characteristics as the first cycle, but the deposition

range was enlarged and the average single layer thickness was thinner.





### 4.3 Case studies

1) Model 1

In order to better display the 3D performance of Sedapp, this paper designed a model called

Model 1. Its length and width ranges were both 200m, and the elevation range was about 10m. The

mesh was 200 × 200 in x-y plane. The time span of the model was set at 10 Ma, and the step size was

set at 0.5 Ma. Sea level was kept constant at 3 m. Its initial topography was set as that in Fig.6a. A river

was set up in the central position of y-axis (y = 100m). The channel shape of the river was set in

advance being a sine curve. Fluvial profile slope is set to a constant of 0.00357, while the sediment

supply rate was not defined since it could vary according to the fluvial profile slope. The other main

parameters of the model are shown in Tab. 1.

The projection of the simulation results on the x-y plane clearly shows the variation

characteristics of the along the shore. When t = 0, the shoreline was a straight line, and the channel was

in the middle of the shoreline. As time went on, the river mouth continued to move forward. From 0 to

2 Ma, the channel first swung to the north, then to the south, and shoreline began to bulge slightly

towards the sea side. From 2Ma, the channel continued to swing southward, until the time approached

4Ma and the river mouth began to turn north slowly. From 4Ma to 6Ma, the channel continued to swing

northward, and the convex part towards the sea side became more and more obvious. From 6Ma to

8Ma, the channel continued the previous trend, while the convex shoreline became asymmetry (an

increasing skewness to the north). From 8 Ma to 10 Ma, the principal line of the channel moved

southward, and the convex shoreline gradually returned to symmetry (Fig.7).

The simulation results also show some interesting features on longitudinal sections. Two sections

(y = 75m and y = 125m) perpendicular to the shoreline direction are selected (see Fig.7f for the

position of the section line). The two sections are located on the north and south sides of the main

channel. The distance between the channel and the two sections is varying. In the southern profile (y =

75m), from 4 Ma to 10 Ma, the isochronous lines of the formation changes from sparse to dense, and

then from dense to sparse (i.e., the thickness of a single clinoform changes from thick to thin first and

then from thin to thick) (Fig.8). This is completely contrary to that observed in the northern profile (y =

125m). From 4Ma to 10Ma, the isochronous lines first changes from dense to sparse, and then from

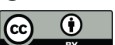



sparse to dense, reflecting that the deposition rate first increases and then decreases (Fig.9).

Under the parameters shown in Tab. 1, due to the existence of estuaries, shoreline will bulge towards the sea side. A closer distance to the river mouth could result in a higher sedimentation rate and a greater shoreline advancing speed. From 2 Ma, the convex shape of the shoreline towards the sea side became more and more apparent, similar to the morphology of some real-world Deltas (Fig.10).

2) Model 2

This code can be applied not only to marginal marine environments but also to the continental fault basins. Taking the 3 + 4 sand groups of the third member of Shahejie Formation in the Gaobei slope belt of Nanpu Sag in Bohai Bay Basin as an example, we conducted a simplified 2D real case study. The basic geological background is as follows: During the deposition period of this set of strata,
the normal fault tectonic movement in the north of the sag was active, which was the main controlling factor leading to the increase of accommodation space. At the same time, the terrigenous clasts came from the north is sufficient, and the basin was in a balanced state (Li et al., 2018). According to the geological background, a simplified reconstruction model (Model 2) was designed, which assumed that the subsidence rate of the boundary fault and sediment supply rate is constant, neglected the effect of
isostasy, and considered the effect of sediment compaction.

The simulation results are shown in Fig.11. From the perspective of temporal and spatial stratigraphy, the shoreline mainly moved towards the sag center during the early stage, and then moved back to the land side. The deepest water depth occurs in the middle south part at 2 Ma (Fig.11a). This shoreline phenomenon is usually called autoretreat (Muto and Steel, 2002). The sand fraction section
shows that the steep slope belt in the north is richer in sand content than the south (Fig.11b). The porosity section shows that the porosity generally decreases from bottom to top. The porosity also varies horizontally, especially when the depth is deeper than 800 m. The porosity in the north is larger than that in the south.

Due to the over-simplified assumptions, the simulation results are not necessarily be consistent
with every practical borehole. However, the general trends are revealed through the simulation, which can strengthen or improve our existing understanding and guide us to seize the main direction. Also, the facies simulation results were in good agreement with the Sedpak results used in Li et al., 2018.



## 5 Discussion

Sedapp is a diffusion-based model, and its transport coefficient is a function of both distance
from estuary and water depth. Compared with most existing diffusion models based only on water
depth, this modification has great advantages in fluvial-deltaic environments, especially for 3D
scenarios. Sedapp not only simulates some surface landscapes, but it also reveals some interesting
internal features. In the sections beside the channel in Model 1, the formation rate of the clinoforms has
close relationship with the distance between the channel and the section. This may be of great
significance to the analysis of ancient strata. Considering the resolution of seismic data, it is easier to
observe the changes in the density of the foreset than to directly find a channel. This may provide some
important supplementary information in areas with less borehole data.

Sedapp also showed strong simulation ability in 2D scenarios. It is not only competent for the
shallow sea environment of continental margin, but also competent for the simulation of continental
fault basin (Fig.11). The simulation results have strong comparability with previous studies (Li et al.,
2018). In addition, Sedapp can avoid some potential problems that the water depth models may meet.
The simulation results of Sedapp and water depth models are not very different where the original slope
is gentle (Fig.12a, Fig.12b). However, when the slope is steep, the differences are obvious: due to the
steep slope and the sharp increase of water depth, the slope break trajectory simulated by the water
depth based model increases significantly, even if the sea level remains unchanged at 6m (Fig.12c).
This is seriously contrary to the common sense, especially in estuary or delta front environments. In
contrast, Sedapp does not face such a problem. As long as the sea level is constant, the slope break line
will remain in a straight line and the clinoforms will also move smoothly to the ocean (Fig.12d).

The transport coefficient is a relatively long-term geomorphologic physical quantity, while wave,
tidal, and current energy are relatively short-term hydrodynamic quantities. However, they are closely
related. A river entering the sea is a type of jet flow phenomenon. The flow velocity decreases rapidly
from the river mouth to the sea, which also has a strong negative correlation with the distance to the
mouth of the river. The contour map of water flow velocity is fan-shaped. At the same time, the
decrease of velocity is also an important cause of sediment deposition, which also explains the close
fan-shaped morphology of a delta front. Correspondingly, an increase in water depth will also decrease
the flow velocity. For the open coast without river injection, a model based on water depth seems to be

reasonable. However, for a coast with river injection, it is difficult to explain the formation of the fan-shaped morphology of a delta. Therefore, it can be concluded that, in more general cases, the transport coefficient should be a function of short-term water energy, which is related to both the

estuary distance and the water depth. When there is river injection, the river process is dominant and the estuary distance function is a reasonable proxy for the transport coefficient. When there is no river injection, the water depth plays the main role. In addition, the particle size is also one of the decisive factors (Nash 1980; Andrews and Bucknam 1987). Hence, a choice function (see Eq. (9)) and differentiated $\alpha$'s are used to adapt different environments and lithologies. Although the current results

of Sedapp seem plausible, these settings for transport coefficient are still empirical. Due to the complex nature of the tranformation from short-term processes to long-term ones, it is difficult to build an accurate bridge between sediment hydrodynamics and stratigraphic formation, while it may be the focus of the next step.

## 6 Code availability

The current version of model is available from the project website:

http://zenodo.org/record/4133262 under the Creative Commons Attribution 4.0 International License.
The exact version of the model used to produce the results used in this paper is archived on Zenodo.
Input data and scripts of the case studies are also presented in this site. For more details about Sedapp,
please contact Jingzhe Li via email lijingzhe@qust.edu.cn.

## 320  Contribution of each author

        JL developed the main algorithm of Sedapp and took the lead in writing the manuscript. PL developed the FVM solver for Sedapp. PL, SS, ZS, YZ, LG, JZ and DD participated in the conceiving of the presented idea. SS supervised the project.

## Acknowledgement

Financial support was provided by the Initial Fund for Young Scholars of Qingdao University of Science and Technology, National Natural Science Foundation of China (42002169) and the Research Funding from King Abdullah University of Science and Technology (KAUST) through the grants



BAS/1/1351-01. Jingfa Li from Beijing Institute of Petrochemical Technology, Jie Chen from Xi'an Jiaotong University and Hua Zhong from Guangdong University of Finance and Economics also

offered constrictive advices.





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



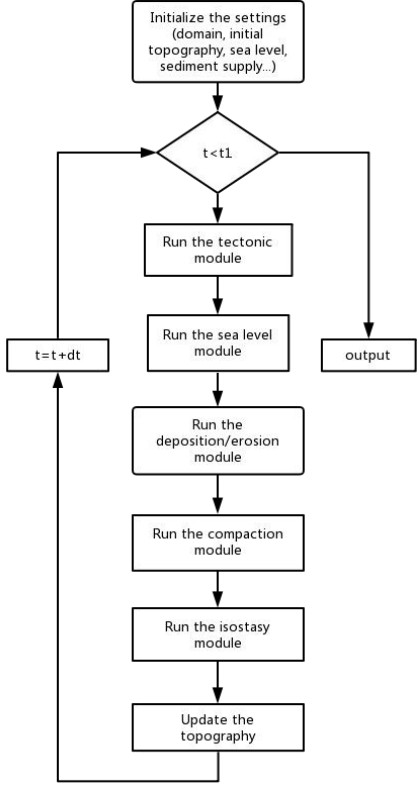

Fig. 1 Flowchart of the algorithms in Sedapp



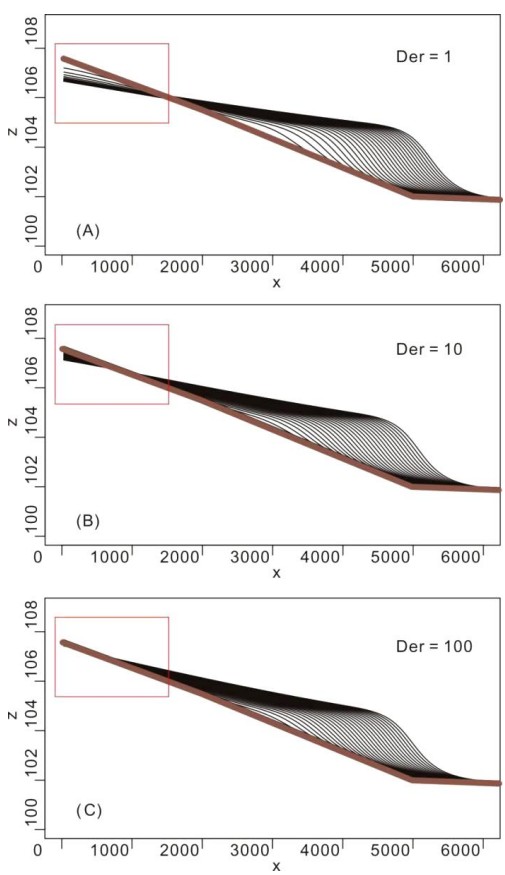

440        Fig. 2 Dip direction section with different Der values (Der = 1, Der = 10, Der = 100 respectively).

Erosion will be switched off if Der is large enough.

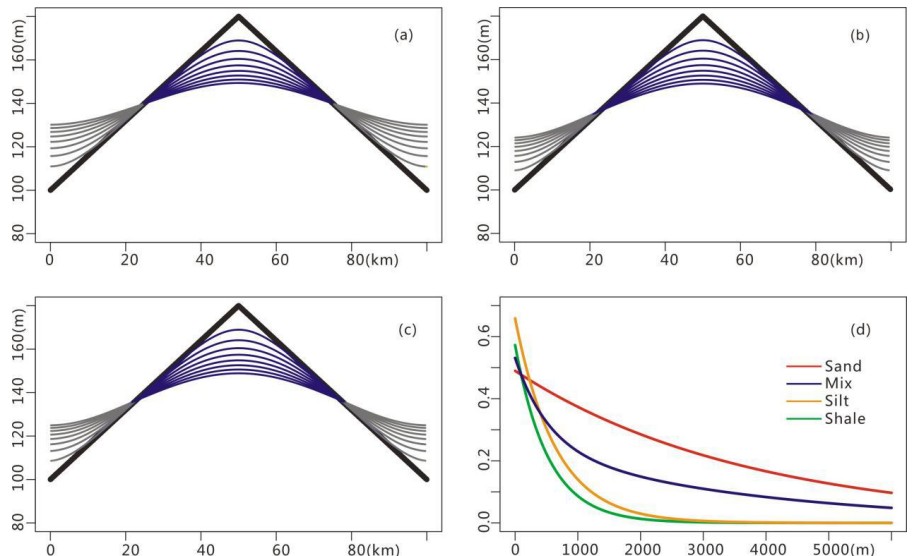

Fig. 3 Customized compaction and the porosity curves. a) the x-z plot with original

depth-porosity scale; b) the x-z plot with magnified depth-porosity scale (x100) to enhance compaction;

445       c) the x-z plot with magnified depth-porosity scale (x1000) to enhance compaction; d) Depth-porosity

curves used in the compaction module (the mix indicates mixed 50%-50% sand and shale. Details see

Athy, 1930; Sclater and Christie 1980)

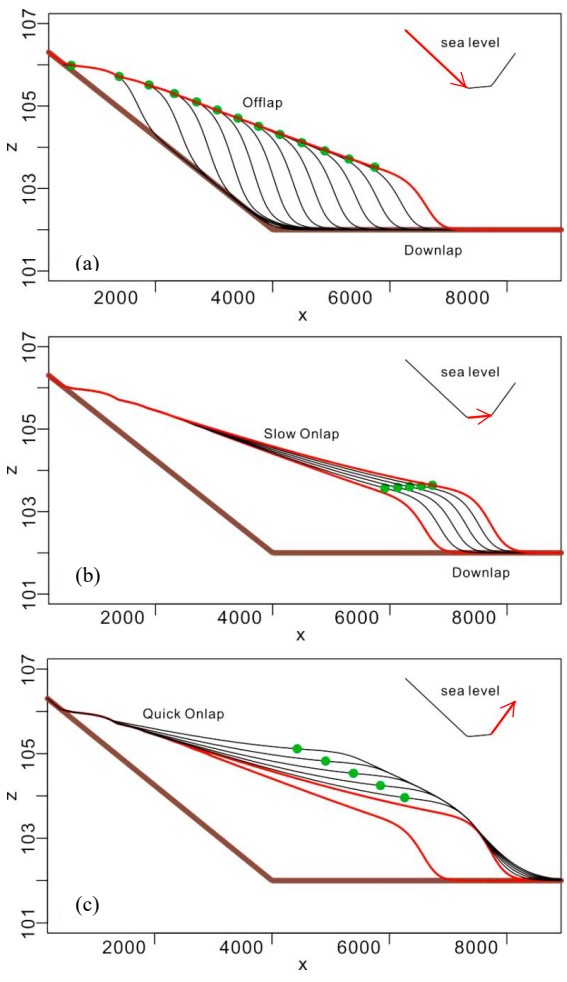

Fig. 4 Typical stacking patterns acquired through different sea level change rates





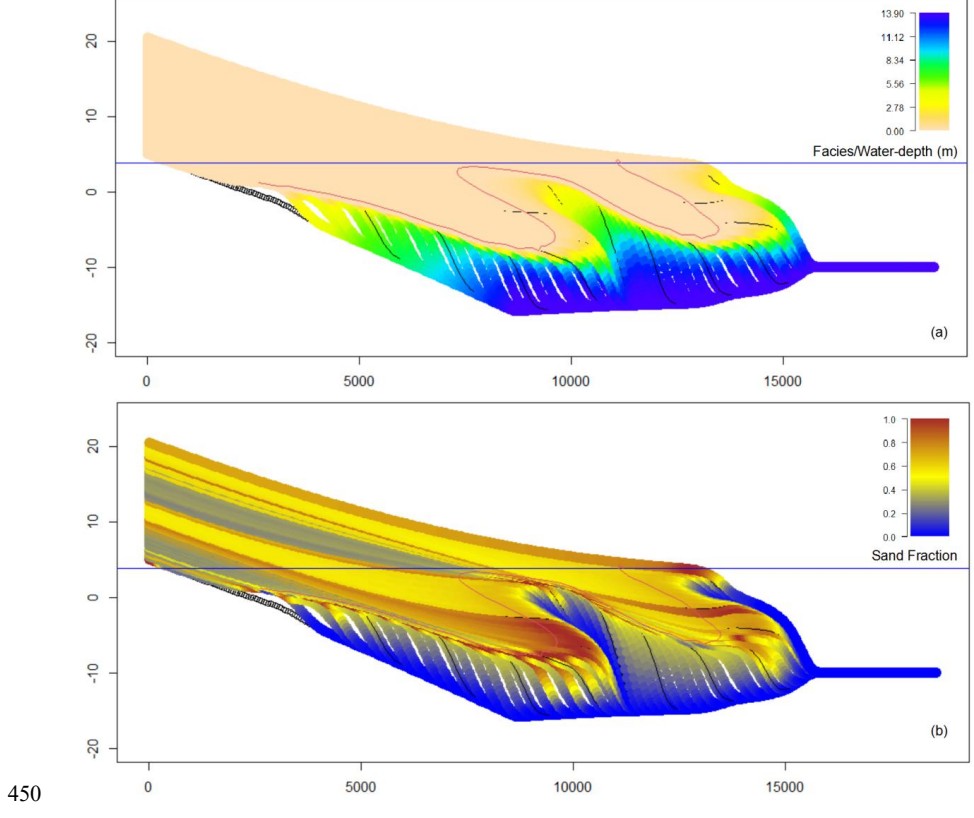


Fig. 5 Simulated stratigraphy under two full sea level cycles. A) facies section and B) lithological

section.



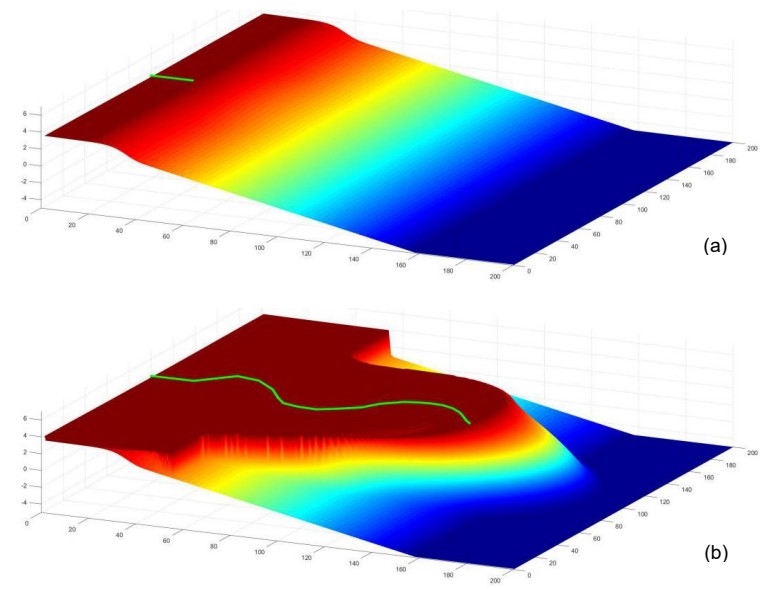


Fig. 6 The initial topography and the simulated results of Model 1. (a): the initial topography; (b):

the topography at t=10 Ma.





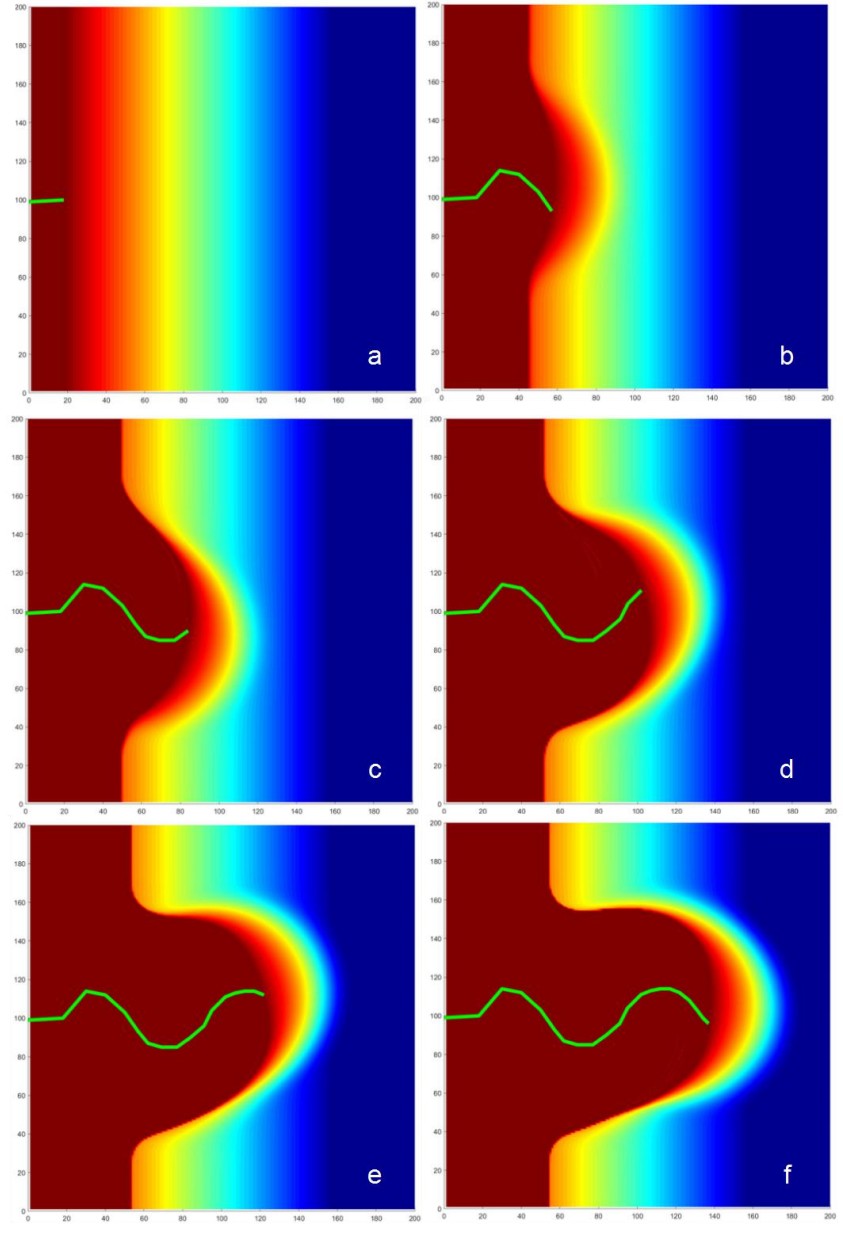

Fig. 7 Plane view of Model 1 results. (a): t=0Ma; (b): t=2 Ma; (c): t=4Ma; (d) t=6Ma; (e) t=8Ma;

460                                                                  (f) t=10Ma.



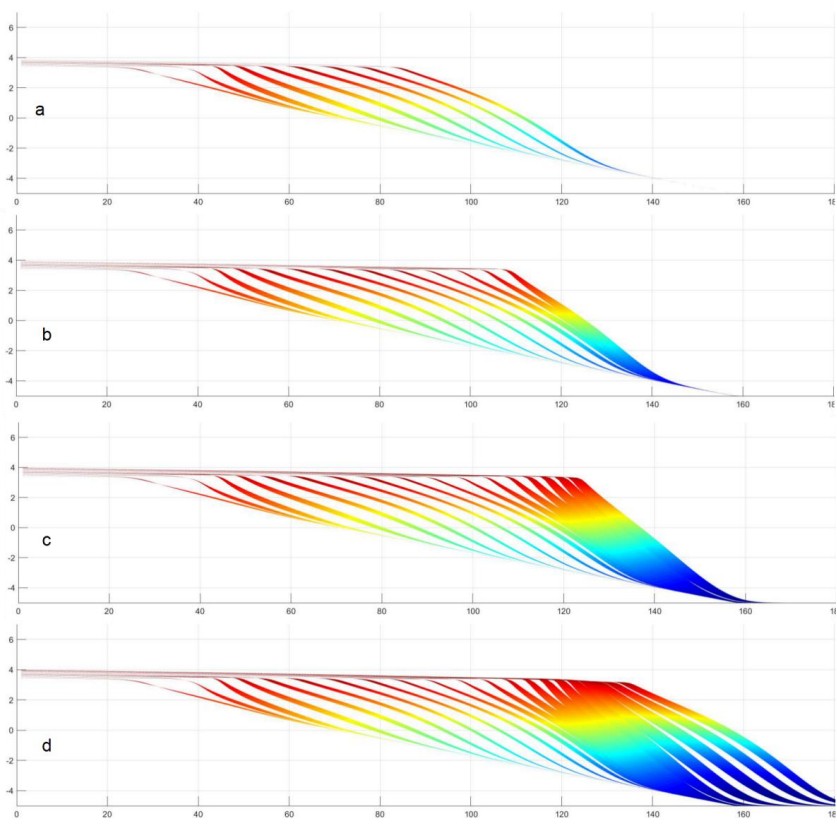

Fig. 8  Cross section at x=75m.    (a): t=4Ma; (b): t=6 Ma; (c): t=8Ma; (d) t=10Ma.



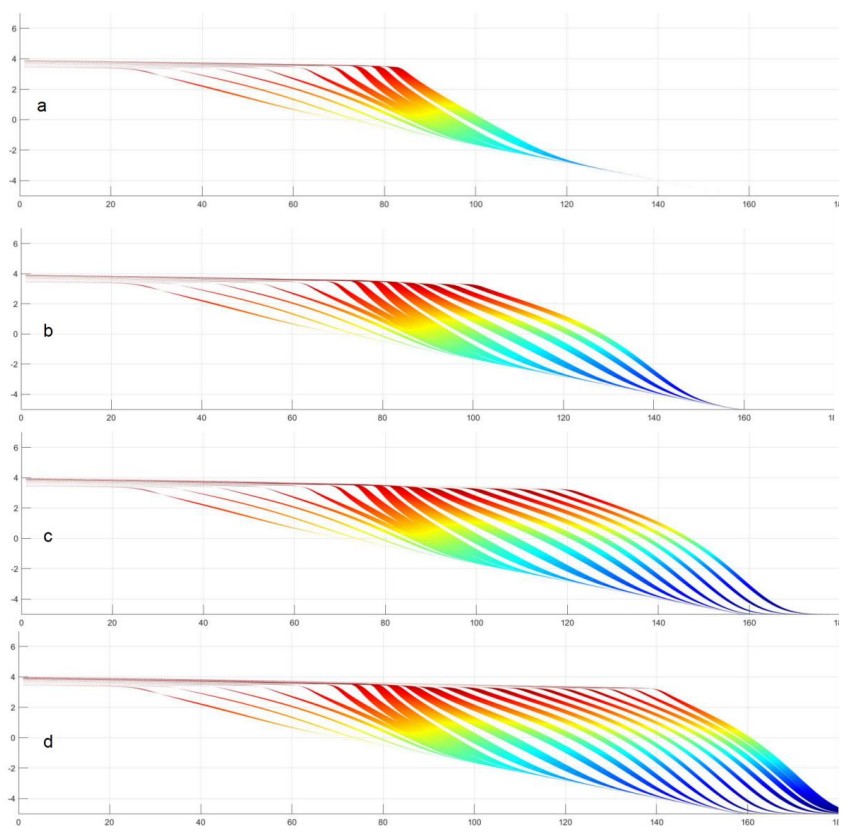

Fig. 9 Cross section at x=125m. (a): t=4Ma; (b): t=6 Ma; (c): t=8Ma; (d) t=10Ma.


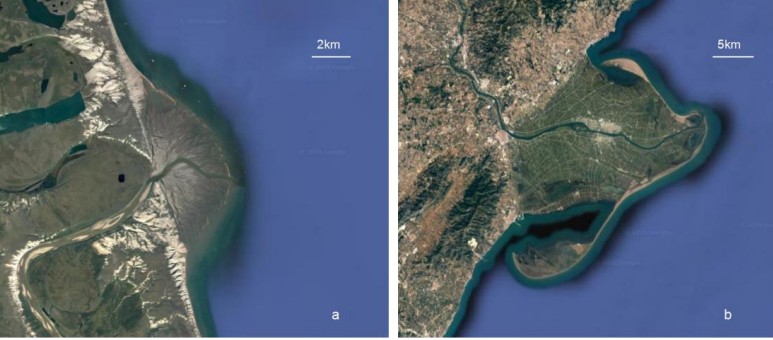

Fig. 10 Horton River Delta in Canada (a) and Ebro Delta in Mediterranean Sea (b) (taken from ©

Google Maps)



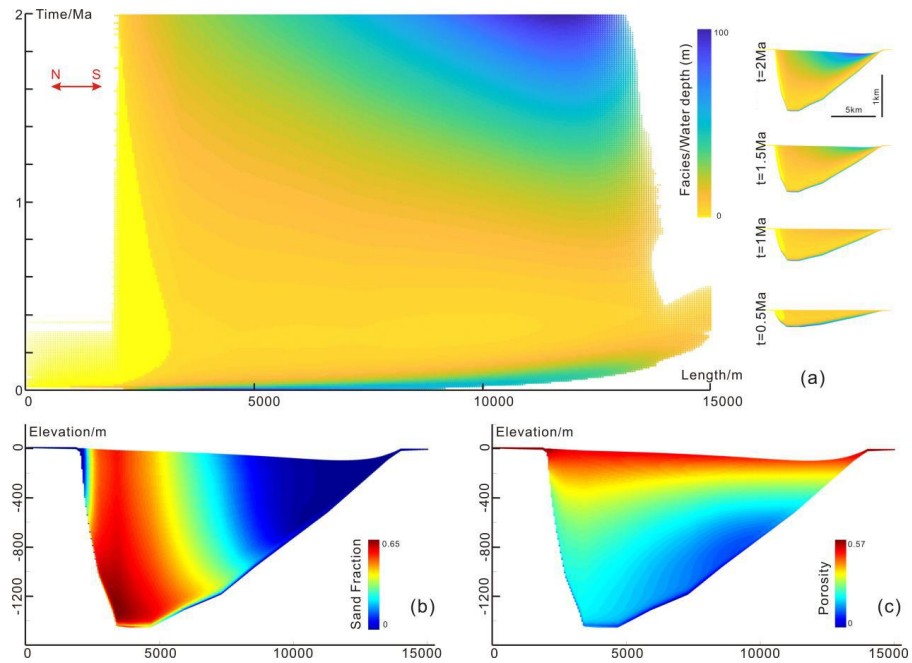

Fig. 11    Simulation results of Gaobei Slope Belt during the study interval. a) Sedapp results of

facies in the time domain (Wheeler diagram) and depth domain at different times; b) Sedapp results of

sand fraction in the depth domain. c) Sedapp results of porosity in the depth domain





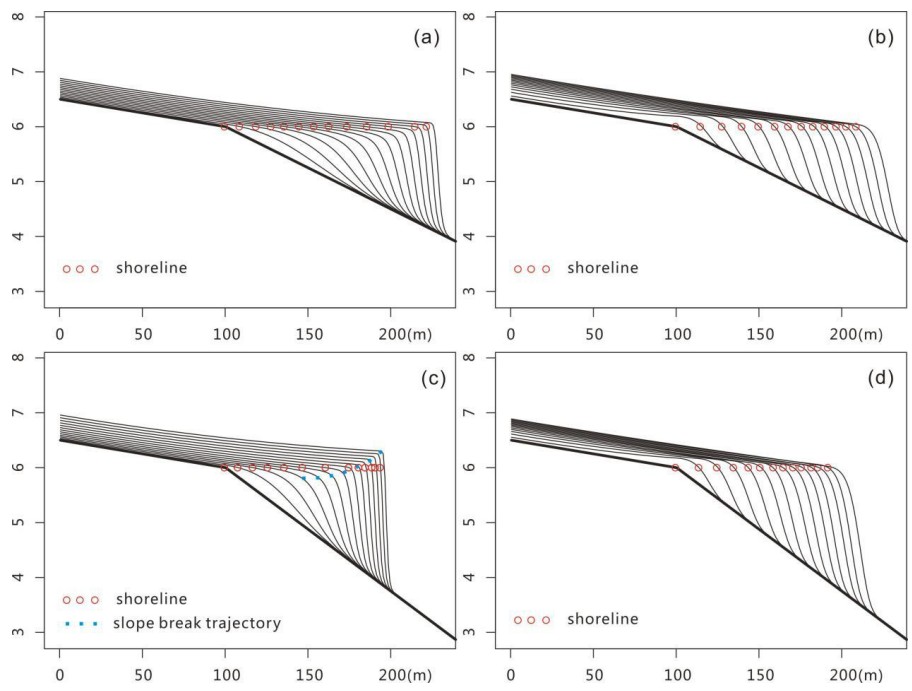

Fig. 12 The differences between two algorithms. a) Clinoforms of gentle slope created in water

depth models; b) Clinoforms of gentle slope created in Sedapp; a) Clinoforms of steep slope created in

water depth models; b) Clinoforms of steep slope created in Sedapp.

Tab. 1 Main simulation parameters of Model1 (see 2.1 above for meanings of the notations)

| Parameter | Value |
|:---:|:---:|
| $\alpha$ | 1000 |
| $\beta$ | 500 |
| $\eta$ | 2 |
| $\alpha_{wd}$ | 10000 |
| $\beta_{wd}$ | 0.16 |
| $\eta_{wd}$ | 1 |
| $\varepsilon$ | 0 |
| $Der$ | 1 |
