# Peer review of "Sedapp v2021: a non-linear diffusion-based forward stratigraphic model for shallow marine environments"

_Geoscientific Model Development, 2020_

## Referee Comment (RC1) · John Armitage (Referee) · 15 Dec 2020

In this manuscript the authors describe a numerical model that transports sediment down slope following a diffusion equation. The model is described as a forward stratigraphic model, as it can simulate the sedimentary deposition within basin settings.

Overall I think the model is OK. It does not break new ground in stratigraphic modelling but it does contain some interesting ideas that might be worthy of publication. The manuscript could however benefit from being restructured to describe better what is new, and to better explain the context of this sort of model.

Major points:

1. The introduction is a bit difficult to follow. The first paragraph introduces sequence

stratigraphy and then ends with the citation of two papers, Burgess, 2012, and Burges and Prince 2015, that discuss how sequence stratigraphy is a paradigm that is no longer fit for purpose. What point are the authors trying to make here? The second paragraph lists different models. The third paragraph defines forward stratigraphic models, but then discusses solvers for Navier Stokes equations (Delft3D). It then discusses "fuzzy logic" and "deductive models" but not in any detail. I think the whole introduction shuld be re-thought-out. Is there to be a discussion of cellular automata models versus models that solve PDEs? Is there to be a discussion of sequence stratigraphy? Ultimately the point should be what does Sedapp advance, but this is missing.

2. What is new about Sedapp? There exist numerical models that solve for sediment transport where the central equation is diffusive. Admittedly the more applied codes are close source and propriety. Is the only advantage of Sedapp that it is opensource? Why did the authors write it in R? Why not use more parallel libraries, see for example eSCAPE developed by Tristain Salles.

3. The functional form of the diffusion coefficient is not justified in any way. Why is it an exponential of distance? Why should erosion be less efficient compared to deposition? In a transport –limited model such as this, I see no argument for defining regions that will be eroded with the "Der" parameter, as erosion is simply a function of the curvature of the topography. I am not understanding something here, sorry.

Other points:

Line 55: Why are these two papers cited for what Is a simple diffusion equation. The equation could be called an Exner equation, and perhaps of a paper is to be cited, Chris Paola's review from the year 2000 would be more appropriate. In general the citations are a bit lazy. It would be worth being more selective about what statement requires a citation, and choosing the relevant citation. For example, pyBadlands (Salles et al., 2018) does not solve for erosion assuming that it transport-limited. It instead solves for

the kinematic wave equation known as the stream power law. Sub marine deposition is a function of slope however. For it to be cited here is out of place.

Equation 1: Has compaction been intentionally left out?

Line 61: The diffusion coefficient can be a function of space and the PDE can still be linear. If however the diffusion coefficient is a function of slope then it would no longer be linear. This statement on this line, "Models with constant $\Gamma$ values are usually called linear models; otherwise, they are known as non-linear models" is not accurate.

Line 65: What is the point of this list? Sedapp is a diffusive model, so it likewise cannot account for mass wasting or biological agents. The next sentences would appear to imply that a non-linear diffusion model can capture these processes, "on the contrary, non-linear models are relatively more flexible". Yet this is not the point, because diffusion models simply cannot capture those processes. Therefore I am confused as to the point the authors are trying to get to.

Line 73 (and in the abstract): The models presented are 2D and 1D. They are not 3D.

Line 81: Linux is likewise free. I don't see an advantage in pushing people to buy a Windows or OS license.

Line 83: "In this paper, we propose a new non-linear model, which is expected to overcome the shortcomings of the existing models". What are the shortcomings of the existing models? They were not explained in the introduction. In fact the "existing models" were not described. What are the existing models?

Line 94: How does a user chose a value for "Der"? Why assume that the diffusion coefficient varies between erosion and deposition? It is the same process that is transporting the sediment therefore I see no argument for why it should change. In the work of Laure Guerit, https://doi.org/10.1130/G46356.1 the transtion from detachment-limited to transport-erosion is discussed in relation to alluvial fan deposition. This reference might provide a starting point for exploring some arguments for the parameter "Der".

Unfortunately, as this function is presented here I find no justification for it.

Equation 4: Why is the diffusion coefficient an exponential function of distance? For sub-aerial sediment transport it would most likely be a function of water flux (see Smith and Bretherton, https://doi.org/10.1029/WR008i006p01506). Water flux does increase down-slope, but why as an exponential function?

Line 104: I have been assuming "D" is the distance down slope, but this might not be the case. "D" is the distance from where?

Code implication: is the code explicit or implicit in time? Have any off-the-shelf solvers been used within the implementation in R? What controls code stability, is there a CFL equivalent?

Line 133: "it will be contrary to the geological knowledge that deposition and erosion processes are two distinct processes with different rates". Who says this? They are both due to the transport of grains of sediment by moving water. The same water.

Case studies: Please confirm that the results are not sensitive to the model resolution. Or if they are explain why.

Line 235: Why does the channel migrate?

Line 291: "This is seriously contrary to the common sense". I think common sense is over rated. Please cite some studies that would suggest that the results of Sedapp are more appropriate.

Code availability: the code comes as a "rar" file. It would be better if the code was hosted on a repository, such as github, gitlab etc, and had a readthedocs with information on how to install and test it.

I found that on my French work laptop running linux I had to remove some non-standard characters within the comments of the code so that Example.r would execute. If this code was on an open repository bugs like this could be reported and fixed with the help

of the community.

I needed some linux libraries to install the library smoothr and to run the code (udunits2-devel on CentOS), which would mean switching to my personal laptop as I don't have sudo privalages on my work workstation. Therefore I switched to a Windows VM to test it. This extra dependence could be signalled in the documentation.

I found that I needed to install "smoothr", "Rcpp", and "Matrix" libraries for the code to run. 3Rcpp" however was not listed in the dependencies. The code however then fails upon trying to create a directory:

38: In dir.create(wdnow) : cannot create dir 'C:\Users\armitagj\Documents\MATLAB\gaobei20', reason 'No such file or directory'

This comes from: Sedapp to be called.R:247: t1p=i*2;wdnow = paste('∼/MATLAB/gaobei',t1p,sep = '');dir.create(wdnow)

With these bugs fixed the code ran on my VM, however given the small processing power the code was very very slow.

The authors could consider creating a Docker container with the code. The point of entry could be a jupyter-notebook that is ready to run Example.r. This would then achieve one of the aims, for a cross-platform model. It would also iron out the small issues I found above.

Summary: Despite all the above issues I think there is something in this work. I find the diffusion equation developed curious and want to understand it better. I therefore hope that my comments are constructive and would be happy to learn more. Furthermore, if the authors do decide to place the code in a repository I can then mark the issues I came up against in getting the code to run.

I hope at the least that this is helpful, John Armitage IFP Energies Nouvelles

---

## Short Comment (SC1) · 21 Dec 2020

Dear authors,

in my role as Executive editor of GMD, I would like to bring to your attention our Editorial version 1.2:

https://www.geosci-model-dev.net/12/2215/2019/

This highlights some requirements of papers published in GMD, which is also available on the GMD website in the 'Manuscript Types' section:

http://www.geoscientific-model-development.net/submission/manuscript_types.html

In particular, please note that for your paper, the following requirement has not been

met in the Discussions paper:

- "The main paper must give the model name and version number (or other unique identifier) in the title."

Please add a version number for Sedapp in the title upon your revised submission to GMD.

Yours,

Astrid Kerkweg
* * *

---

## Referee Comment (RC2) · Anonymous Referee #2 · 26 Jan 2021

This work presented the features and application of Sedapp, which is a non-linear open-source R code for Forward Stratigraphic Modelling. The manuscript is generally well written and structured, and the authors did show some interesting ideas of the model. In my opinion, the manuscript is suitable for publication in GMD, after the authors have addressed the following comments and questions: 1. The authors said "Although many advances have been made in the field of forward stratigraphic modelling (FSM), there are still some shortcomings to the existing models." in the abstract. While I did not find the detailed description in the introduction part. Please modify or rewrite the related part in the manuscript. 2. As the authors said in the text that many existing models are not open to the public. The open-source feature of Sedapp is apparently a good aspect for its availability. There are actually many alternative programming languages that can satisfy this feature. For example, some other open source models were written in Fortran, C or python. Why did the authors write Sedapp in R? 3. Fig. 7: This figure is interesting whose subplots show different stages of a river-dominated delta. I also noticed that the river is of a curvy shape. Why was the channel shifting like this? Is there any forces made it so? By the way, the initial segment is not very smooth, Is there any special requirement for that? Or the authors could modify it and make it look better. 4. The Der settings in the model is very interesting, which could distinguish the depositional and erosional two processes. But what geological knowledge support this setting? Also, which variable name in the code corresponds to it? 5. In the code, I noticed a parameter called isostasy. With the default setting, I could get the same figure as in the text. However, when I switched the value, the results became very different. The left part dropped far below the expected place, while the right part also changed a lot. I think the authors should make a brief introduction in the documents about these parameters. 6. Porosity of the strata successions is a very import parameter concerned by resource geologists. The porosity changes at every moment as long as the overlying strata was changed, so does the previously deposited strata. That means whenever a new layer is generated, porosity of both new and old layers would be updated. I wonder how is this process implemented in the code. 7. When I ran the "example.r", I also noticed that, the "fluvial slope" is updated at each step. Like: "Fluvial slope is ...", "Slope angle is ...°". What is it? Why is it needed to be monitored here? 8. For the comments of the code, I think the authors should improve them. There are many subroutine files to the main program. However, the comment styles within these files vary from each other. The authors should put them into a uniform manner.

---

## Author Comment (AC1) · 17 Feb 2021

Thank you very much for your comments. Here below is our reply: Comment 1: Please add a version number for Sedapp in the title upon your revised submission to GMD. R1: We have modified the title as suggested.

---

## Author Comment (AC2) · 17 Feb 2021

Dear referee 2, Thank you very much for your careful read and helpful comments. Here below is our reply:

Comment 1: The authors said "Although many advances have been made in the field of forward stratigraphic modelling (FSM), there are still some shortcomings to the existing models." in the abstract. While I did not find the detailed description in the introduction part. Please modify or rewrite the related part in the manuscript. R1: This part has been re-written as suggested in the new version of the manuscript.

Comment 2: As the authors said in the text that many existing models are not open to the public. The open-source feature of Sedapp is apparently a good aspect for

its availability. There are actually many alternative programming languages that can satisfy this feature. For example, some other open source models were written in Fortran, C or python. Why did the authors write Sedapp in R? R2: We write the code in R firstly because it is easier to use and easier to make open-source. Also, there is an off-the-shelf FVM solver written in R. For the sake of our research compatibility, we chose to develop the code based on it.

Comment 3: Fig. 7: This figure is interesting whose subplots show different stages of a river-dominated delta. I also noticed that the river is of a curvy shape. Why was the channel shifting like this? Is there any forces made it so? By the way, the initial segment is not very smooth, Is there any special requirement for that? Or the authors could modify it and make it look better. R3: The shape of the river is a kind of predefined information. It is mainly to facilitate to show the influence of a curved channel on the depositional results. The initial segment of these channels has been modified.

Comment 4: The Der settings in the model are very interesting, which could distinguish the depositional and erosional two processes. But what geological knowledge support this setting? Also, which variable name in the code corresponds to it? R4: The introduction of Der is generally to facilitate the result-fitting in downstream and subsequent shallow marine areas for some complex situations. For example, some initial surface is "hardground", which is very difficult to be eroded. While the overlying deposition process is relatively easy. The value of Der is usually empirically defined and modified based on the stratigraphic record. In the code, this parameter is called "dep.ero.ratio" in the main program.

Comment 5: In the code, I noticed a parameter called isostasy. With the default setting, I could get the same figure as in the text. However, when I switched the value, the results became very different. The left part dropped far below the expected place, while the right part also changed a lot. I think the authors should make a brief introduction in the documents about these parameters. R5: Isostasy is the state of gravitational equilibrium between Earth's crust (or lithosphere) and mantle such that the crust "floats" at

an elevation that depends on its thickness and density. A brief introduction about this has been added in the new code documentation as suggested.

Comment 6: Porosity of the strata successions is a very important parameter concerned by resource geologists. The porosity changes at every moment as long as the overlying strata was changed, so does the previously deposited strata. That means whenever a new layer is generated, porosity of both new and old layers would be updated. I wonder how is this process implemented in the code. R6: We use porosity functions to implement the compaction process. The current functions are shown in "phi.r" file of the code. Every previously formed layer is also updated when a new layer is formed with a nested loop. Details can be seen in the "Compaction Process" module of the "Sedapp to be called.R" file.

Comment 7: When I ran the "example.r", I also noticed that, the "fluvial slope" is updated at each step. Like: "Fluvial slope is ...", "Slope angle is ...". What is it? Why is it needed to be monitored here? R7: These angles are monitored because the fluvial slope is a very important parameter for the depositional processes. Also, this is a parameter that is very easy to be measured in modern sedimentary counter-parts. This could provide us with a good opportunity to test the rationality of the simulation.

Comment 8: For the comments of the code, I think the authors should improve them. There are many subroutine files to the main program. However, the comment styles within these files vary from each other. The authors should put them into a uniform manner. R8: These files have been modified as suggested.

---

## Author Comment (AC3) · 19 Feb 2021

Dear John, Thank you very much for your constructive and heuristic reviewing work. Here below is our reply to the comments:

Major points: Comment 1: The introduction is a bit difficult to follow. The first paragraph introduces sequence stratigraphy and then ends with the citation of two papers, Burgess, 2012, and Burges and Prince 2015, that discuss how sequence stratigraphy is a paradigm that is no longer fit for purpose. What point are the authors trying to make here? The second paragraph lists different models. The third paragraph defines forward stratigraphic models, but then discusses solvers for Navier Stokes equations (Delft3D). It then discusses "fuzzy logic" and "deductive models" but not in any detail.

[Figure]

I think the whole introduction shuld be re-thought-out. Is there to be a discussion of cellular automata models versus models that solve PDEs? Is there to be a discussion of sequence stratigraphy? Ultimately the point should be what does Sedapp advance, but this is missing. R1: The related parts have been re-written as suggested.

Comment 2: What is new about Sedapp? There exist numerical models that solve for sediment transport where the central equation is diffusive. Admittedly the more applied codes are close source and propriety. Is the only advantage of Sedapp that it is opensource? Why did the authors write it in R? Why not use more parallel libraries, see for example eSCAPE developed by Tristain Salles. R2: The functional form of the diffusion coefficient is an exponential of distance. This is different from many existing models whose coefficients are water-depth related. As shown in the discussion part, this modification has advantages in both the stability of the slope break trajectory and the controllability of the fluvial-deltaic shoreline shape . We write the code in R, not only because it is easier to make open-source, but also that there is an off-the-shelf FVM solver written in R. For the sake of our research compatibility, we chose to develop the code based on it. The referee's suggestion is a very good alternative, and we may try it in future studies.

Comment 3: The functional form of the diffusion coefficient is not justified in any way. Why is it an exponential of distance? Why should erosion be less efficient compared to deposition? In a transport –limited model such as this, I see no argument for defining regions that will be eroded with the "Der" parameter, as erosion is simply a function of the curvature of the topography. R3: The functional form of the diffusion coefficient was assumed as an exponential of the distance from the estuary. This is based on some existing model assumptions and several jet flow experiments. Generally, the diffusion coefficient is assumed to fall exponentially with the water depth (e.g. Syvitski and Hutton, 2001). This may be to reflect the relationship between short-term hydrodynamic energy and long-term geomorphological processes. The hydrodynamic energy decreases with the increase of the water depth. While the hydrodynamic energy also

shows a similar changing rule with the distance from the jet outlet as revealed by jet flow experiments. Based on this fact, we designed the current form of the diffusion coefficient. In practical application, the modified model indeed made plausible results, which have high similarity with those of the existing related models. Additionally, this model has higher flexibility against the existing ones when handling river inject problems. The essence behind this phenomenon is indeed a problem worthy of study in the future, which may bridge short-term hydraulic processes and long-term geomorphological processes. The erosion is not necessarily less efficient compared to deposition. The introduction of Der is mainly to facilitate some complex situations. For example, for a given location, some bed surface is "hardground", which is very difficult to be eroded. While the overlying deposition process is relatively easy. In this case, the distinction seems necessary. For a long-term stratigraphic forming process, there may exist many sedimentary discontinuities, which may provide long enough time to generate some "hardgrounds". The Der is just used to offer one more choice.

Other points: Comment 4: Line 55: Why are these two papers cited for what Is a simple diffusion equation. The equation could be called an Exner equation, and perhaps of a paper is to be cited, Chris Paola's review from the year 2000 would be more appropriate. In general the citations are a bit lazy. It would be worth being more selective about what statement requires a citation, and choosing the relevant citation. For example, pyBadlands (Salles et al., 2018) does not solve for erosion assuming that it transport-limited. It instead solves for the kinematic wave equation known as the stream power law. Sub marine deposition is a function of slope however. For it to be cited here is out of place. R4: The citations have been modified as suggested.

Comment 5: Equation 1: Has compaction been intentionally left out? R5: Eq. (1) here is an original equation, which is to explain the general form of such kind of models. The code implement actually includes the compaction process.

Comment 6: Line 61: The diffusion coefficient can be a function of space and the PDE can still be linear. If however the diffusion coefficient is a function of slope then it would

no longer be linear. This statement on this line, "Models with constant $\Gamma$ values are usually called linear models; otherwise, they are known as non-linear models" is not accurate. R6: The original statement was indeed not accurate, it has been modified as suggested in the text. The corresponding parts of Eq. (4) have also been modified.

Comment 7: Line 65: What is the point of this list? Sedapp is a diffusive model, so it likewise cannot account for mass wasting or biological agents. The next sentences would appear to imply that a non-linear diffusion model can capture these processes, "on the contrary, non-linear models are relatively more flexible". Yet this is not the point, because diffusion models simply cannot capture those processes. Therefore I am confused as to the point the authors are trying to get to. R7: The original expressions are indeed not very appropriate, and they have been modified as suggested in the text.

Comment 8: Line 73 (and in the abstract): The models presented are 2D and 1D. They are not 3D. R8: These places have been modified as suggested.

Comment 9: Line 81: Linux is likewise free. I don't see an advantage in pushing people to buy a Windows or OS license. R9: Linux is indeed a better choice in regards of accessibility. While when we developed Sedapp, the majority of our team were using Windows. For the sake of convenience, we made it on Windows. However, R is a cross-platform language. The core codes could also be run on Linux. In the next stage, we will replace the windows-only parts with some full platform ones in order to make it truly cross-platform.

Comment 10: Line 83: "In this paper, we propose a new non-linear model, which is expected to overcome the shortcomings of the existing models". What are the shortcomings of the existing models? They were not explained in the introduction. In fact the "existing models" were not described. What are the existing models? R10: The existing models here indicate the ones with diffusion coefficients that are only water-depth related. For example, the coefficient was assumed to fall exponentially with the water depth (e.g. Syvitski and Hutton, 2001). The main shortcomings of these coefficients are at the instability of the slope break trajectory and the poor controllability of the fluvial-deltaic shape along the shore. This part has been modified and detailedly explained in the new introduction.

Comment 11: Line 94: How does a user chose a value for "Der"? Why assume that the diffusion coefficient varies between erosion and deposition? It is the same process that is transporting the sediment therefore I see no argument for why it should change. In the work of Laure Guerit, https://doi.org/10.1130/G46356.1 the transition from detachment-limited to transport-erosion is discussed in relation to alluvial fan deposition. This reference might provide a starting point for exploring some arguments for the parameter "Der". Unfortunately, as this function is presented here I find no justification for it. R11: We have carefully read the work of Laure Guerit et al. (2019) as suggested, which does provide a lot of inspiration. While it includes more about the material flux in the whole catchment-fan regions. It is a bit beyond the scope of the current study. The main scope of Sedapp is focused on the area available for sediment accumulation in the downstream and subsequent shallow marine regions (the 3rd of the three factors for sediment record in Armitage et al., 2011). The introduction of Der is generally to facilitate the result-fitting in the above areas for some complex situations. For example, some initial surface is "hardground", which is very difficult to be eroded. While the overlying deposition process is relatively easy. The value of Der is usually empirically defined and modified based on the stratigraphic record.

Comment 12: Equation 4: Why is the diffusion coefficient an exponential function of distance? For sub-aerial sediment transport it would most likely be a function of water flux (see Smith and Bretherton, https://doi.org/10.1029/WR008i006p01506). Water flux does increase down-slope, but why as an exponential function? R12: The functional form of the diffusion coefficient was assumed as an exponential of the distance from the estuary. However, as mentioned in Line 105, this assumption works only in the marine portion. Instead, as shown in Eq. (4), the water depth in sub-aerial portion is actually 0, where the coefficient generally reduces to a user-defined parameter. The paper

(Smith and Bretherton, 1972) provides very useful inspirations for the improvement of the code for the sub-aerial portion. We may take the mass flux of the entire source-sink system into consideration in future versions of the code.

Comment 13: Line 104: I have been assuming "D" is the distance down slope, but this might not be the case. "D" is the distance from where? R13: D is the distance from the river mouth. It works only in the marine portion.

Comment 14: Code implication: is the code explicit or implicit in time? Have any off-the-shelf solvers been used within the implementation in R? What controls code stability, is there a CFL equivalent? R14: The code is implicit in time. There is an off-the-shelf FVM solver used in it. The time step and mesh size used in the simulation satisfy the CFL condition, which guarantees the code stability.

Comment 15: Line 133: "it will be contrary to the geological knowledge that deposition and erosion processes are two distinct processes with different rates". Who says this? They are both due to the transport of grains of sediment by moving water. The same water. R15: The erosion rate is not necessarily different from the deposition rate. What we wanted to say is about some special situations. For example, for a given location, some initial surface is "hardground", which is very difficult to be eroded. While the overlying deposition process is relatively easy. For a long-term stratigraphic forming process, there may exist many sedimentary discontinuities, which may provide long enough time to generate some "hardgrounds". In this case, the distinction seems reasonable. There are indeed some ambiguities in this original expression, and we have modified this sentence in the text.

Comment 16: Case studies: Please confirm that the results are not sensitive to the model resolution. Or if they are explain why. R16: The results are not sensitive to the model resolution.

Comment 17: Line 235: Why does the channel migrate? R17: The shape of the river is a kind of predefined information. It is mainly used to reflect the influence of a curved

channel on the depositional results.

Comment 18: Line 291: "This is seriously contrary to the common sense". I think common sense is over rated. Please cite some studies that would suggest that the results of Sedapp are more appropriate. R18: This sentence has been modified as suggested in the text.

Comment 19: Code availability: the code comes as a "rar" file. It would be better if the code was hosted on a repository, such as github, gitlab etc, and had a readthedocs with information on how to install and test it. R19: The modified new version will be uploaded to both Github and Zenodo. An installation instruction has been added in the documentation as suggested.

Comment 20: I found that on my French work laptop running linux I had to remove some non-standard characters within the comments of the code so that Example.r would execute. If this code was on an open repository bugs like this could be reported and fixed with the help of the community. R20: These comments of code have been modified as suggested in the new version of the code.

Comment 21: I needed some linux libraries to install the library smoothr and to run the code (udunits2- devel on CentOS), which would mean switching to my personal laptop as I don't have sudo privalages on my work workstation. Therefore I switched to a Windows VM to test it. This extra dependence could be signalled in the documentation. I found that I needed to install "smoothr", "Rcpp", and "Matrix" libraries for the code to run. 3Rcpp" however was not listed in the dependencies. R21: These extra dependencies have been signalled as suggested in the new version of the documentation.

Comment 22: The code however then fails upon trying to create a directory: 38: In dir.create(wdnow) : cannot create dir 'C: Users armitagj Documents MATLAB gaobei20', reason 'No such file or directory' This comes from: Sedapp to be called.R:247: t1p=i*2;wdnow = paste('âĹij/MATLAB/gaobei',t1p,sep = ");dir.create(wdnow). With these bugs fixed the code ran on my VM, however given the
small processing power the code was very very slow. The authors could consider creating a Docker container with the code. The point of entry could be a jupyter-notebook that is ready to run Example.r. This would then achieve one of the aims, for a cross-platform model. It would also iron out the small issues I found above. R22: These bugs above have been corrected in the code. A Jupiter-notebook is indeed a better tool for the cross-platform realization. However, there is an off-the-shelf FVM solver needs to be loaded before the initialization. Therefore, it is not more convenient in this way. While we are trying to slim the solver in order to make it available in the potential Jupiter-notebook version.

---

## Author Response (AR2)

**Author's response**

**Reviewer #3**

Major points on the paper:

- Line 47: Why "especially in the shallow marine environments"? There are several published papers about Forward Stratigraphic Modeling in continental and deep marine environments that attest to the applicability of this method in a variety of sedimentary environments. I think it would be better to cut off this statement to avoid the risk of misleading the reader to believe that Forward Stratigraphic Modeling is a method limited to shallow marine environments.
Answer: This place has been modified as suggested.

- Line 72: It is unclear to me what the authors meant by "fractal stratigraphic theory" and, unfortunately, I do not have access to the book they cited. Is it somehow related to Schlager's (2004) scale-invariant fractal model for sequence stratigraphy? If so, I am not sure what would be the use of this citation here in reference to the formation of hardgrounds. I suggest rewriting that sentence to make clear what kind of hardgrounds the authors want to mention or finding a better example for the formation of strata resistant to erosion.
Answer: The fractal stratigraphic theory of Miall (2015) does have some relationships with that of Schlager (2004), while their focuses are different. The fractal conception in Miall (2015) was mainly used to explain the predominance of missing time in the sedimentary record. This indicates that the time gap between two adjacent small layers can be long enough that the underlying layer could become "hard" before the overlying layer starts to form. The "hardground" here means the loose sediments become solid sedimentary rocks due to post sedimentation processes. In order to make it clear, this sentence has been modified in the text.

- Lines 72 and 73: By "This is actually a reflection of the efficiency ratio of deposition to erosion. This is less involved in the existing FSM models." seem to imply that the existing FSM models do not take into consideration the dynamic between deposition and erosion. This would be simply not true as the dynamic between deposition vs. erosion according to the efficiency of sediment transport and sediment supply is part of the fundamental principles of some of the most used FSM tools. I think the authors should rephrase this part to make clear what they mean by "efficiency ratio of deposition to erosion" if they really think that it is something missing in other FSM tools.
Answer: This place has been modified as suggested.

- Line 115: What the authors mean by "hydraulic characteristic energy" in terms of oceanographic and sedimentological processes? And how does it differ from the environment energy represented by ε? I think it is important to mention what exactly E and ε were designed to simulate.

Answer: For E in Eq. 9, it is a denominator of an exponent. The corresponding numerator is a transformed distance term, which could be regarded as a proxy to the river injection related hydraulic energy as discussed in the text. In order to make the exponent dimensionless, the denominator and the numerator should have the same dimension. Thus, E was considered as "hydraulic characteristic energy". While for ε, it does not contribute very much to the deposition geometry around the shoreline. Instead, it could significantly affect the sedimentation deep in the basin. This seems to have little thing to do with the transportation capability originated from the river injection. It's more like the energy inherent in the basin that carries the sediments to the basin center. An increasing ε usually leads to an increasing amount of deposition in the interior basin. Thus, ε was considered as the inherent environment energy in the basin.

- Line 133: How exactly the model differentiates "hardground" from easily erodible layers? Is it somehow user-defined for cells in the model? Or is it automatically calculated for each cell in the model? If so, how is it calculated for each cell? In subtopic 3.2 the authors mention that a variety of lithological and environmental factors affect erosion rate in Sedapp v2021, but how exactly these factors are taken into consideration? As the "efficiency ratio of deposition to erosion" was mentioned as an innovative aspect of Sedapp v2021, I think it is important to understand how all parameters related to erosion work in Sedapp.
Answer: It is user-defined. As mentioned in part 3.2, many factors could affect the erosion rate. For ancient strata, it is usually very difficult to quantify the rate through the analysis of the factors in geologic time. Instead, the rate can be defined empirically and adjusted through result fitting. Thus, Sedapp just provides this function for the users to easily customize the value.

- Line 183: The proportions of sediment classes were changed between figures 3a, 3b, and 3c? I think it would be better to keep the same sediment proportions in a, b, and c to illustrate the effect of different depth-porosity curves alone. Might also be interesting to show the effect in compaction of varying the sediment proportions and keeping the same depth-porosity curve. But if for the sake of simplicity, the authors prefer to only show an example where both the depth-porosity curves and sediment proportions change, they must indicate in the figure legend what is the sediment proportion variation between 3a, 3b, and 3c.
Answer: No, it's not. The proportions of sediment classes were kept constant in figures 3a, 3b, and 3c (50% sand and 50% mud). Fig. 3 is used to show the compaction effect in Sedapp. The difference among these 3 sub-figures is the depth scale. Fig. 3a shows the original compaction effect, while 3b and 3c show the magnified compaction effects by scaling depths.

- Line 219: I believe there is a missing reference to figure 7 by the end of this phrase.
Answer: This place has been modified as suggested.

- Line 221: The way geological time is counted in this paragraph and in figure 7 is very confusing from the point of view of a geologist. In geology, time is always counted backward as "years ago" or "years before present". For geologists "Ma" always means "millions of years ago". Sometimes "Myr" is used to indicate a time span in million years from one moment to another, rather than the "Ma" that is always in reference to the present day. So, in this paragraph and figure 7, it is like all ages are inverted from a geological point of view. I suggest using "Myr" instead of "Ma" if the author prefer to count time forwardly despite of the geological nature of the paper or even better invert all ages and use: t = 0 (10 Ma); t = 4 (8 Ma); t = 8 (6 Ma); t = 12 (4 Ma); t = 18 (2 Ma) and t = 20 (0 Ma). The same is valid for all ages in the following figures and paragraphs mentioning geological time.

Answer: All these places have been modified as suggested.

- Line 229: I believe there is a missing reference to figure 8 by the end of this phrase.

Answer: This place has been modified as suggested.

- Line 277: I suppose there are things that Sedapp v2021 do better than the Sedpak model used by Li et al., 2018, hence the reason to publish this paper. How do Model 2 presented in this paper compare to the model presented by Li et al., 2018? What is the advantage of using Sedapp v2021? I think there is a good opportunity here to clearly state the contribution of Sedapp v2021 to Forward Stratigraphic Modeling. Please elaborate more.

Answer: Sedpak is a good tool in basin filling simulation. In the case of Gaobei fault basin, both Sedapp and Sedpak can perform well. As for the advantage, I think the physical meaning of the parameters in Sedapp is more clear. In addition, the graphic function in Sedapp is more flexible. For example, sand fraction plots can be easily created by Sedapp, while I didn't find the corresponding function in Sedpak. The main purpose here is to reflect that Sedapp can do a good job in the simulation of continental fault basins, which can be comparable to the relative mature Sedpak.

- Lines 278 to 282: This part seems much more like results that should be presented in the previous section than Discussion. All models shown in figure 12 were made using Sedapp v2021? If so, this part of the text and figure 12 must be moved to the "Verification of Sedapp" section. Furthermore, it must be stated how exactly model 12a differs from 12b and model 12c differs from 12d. Is it only a matter of turning on and off the diffusive transport as a function of distance to shore? Or these models differ between them in another way?

Answer: Yes, all these models shown in Fig. 12 were made using Sedapp. The only difference among these models is the choice of the transport coefficient algorithm. The example here is mainly used to show that Sedapp can avoid some potential problems that water depth models may not overcome. While in order make it more clear, some modifications have been made in the text as well as the caption of Fig. 12.

- Lines 282 to 284: The statement about the preservation of shoreface profile angle seems misplaced in this paper. To the extent of my knowledge, the preservation of shoreface profile is always discussed in the context of the action of processes that do not seem to be simulated in Sedapp v2021, such as sediment reworking above the wave base and sediment collapse when a clinoform reaches a critical slope. There are plenty of examples in the literature about varying shoreface profiles during regressive system tracts when processes like these do not affect sediment deposition and preservation significantly. I do not see the point in discussing the preservation of shoreface profile if the processes that allow it are not taken into consideration individually by the models presented in this paper. Suffice to point out that in figure 12d the sigmoidal shape of the deltaic layers persists during the progradation of subsequent stratigraphic layers and that the slope break is never located landward of the shoreline (which is the true issue in figure 12c).
Answer: This place has been modified as suggested.

- Line 352: Missing year of publication?
Answer: The year has been added as suggested.

- Figure 7: What are the min and max values of the color scale? It should be at least mentioned in the figure's legend if not plotted on the figure.
Answer: The max and min values of the color scale have been added on the figure as suggested.

Suggestions about minor points:
– Line 31: The word "archive" is used three times in the first paragraph to describe sedimentary deposits. I find the use of this word very unusual in this context. The meaning is clear, so I suppose that there is no problem in using the word "archive" there, but I would rather say something like: The sedimentary successions formed in these areas are an important record of the past interactions. In addition, shallow marine stratigraphic record itself can be an ideal hydrocarbon accumulation place. From this record, many theoretical and field studies have made great achievements and accumulated a wealth of data in the past decades.
Answer: These places have been modified as suggested.

– Line 43: I think there is a missing reference here. In my opinion, it would be appropriate to include here at least one citation about Dionisos. It is free for academic use and it pioneered the field of Forward Stratigraphic Modeling with a considerable contribution through dozens of published scientific works during the last 20 years. I suggest adding to the references either Granjeon and Joseph (1999) or Granjeon (2014) as these papers are more methodological.
Answer: This place has been modified as suggested.

– Line 76: This phrase seems a little too bold to me. Forward Stratigraphic Modeling has been evolving significantly in the last two decades, but it is true that the existing FSM tools still have many shortcomings. That said, although Sedapp v2021 appears to propose some clever solutions, it surely will not overcome all shortcomings of the existing models. I think "… is expected to overcome some of the shortcomings of the existing models" would be more appropriate.

Answer: This place has been modified as suggested.

– Lines 86; 104: I suppose the authors meant "class of sediment" instead of "class of lithology".

Answer: This place has been modified as suggested.

– Line 153: What is "c"? I suppose c is the variable used in equation 9 in the previous topic to differentiate the characteristics of different sediment types, but this chapter is too far from the last mention of c. I had to go back in the text to understand what it was. I think it would be a good idea to reference equation 9 here to help a reader like me find where c was described.

Answer: A reference has been added for "c" as suggested.

– Line 160: I think "underlying strata" would be more appropriate than "lower strata".

Answer: This place has been modified as suggested.

– Line 206: It would be nice to add a line in figures 5a and 5b representing the boundary between these two cycles as a visual reference for readers that are not used to analyze this kind of results. It would make it easier to identify the aspects of figure 5 mentioned in this paragraph.

Answer: The boundary lines have been added as suggested.

– Line 263: What do the authors mean by "guide us to keep the general direction"? Validate the previously proposed conceptual model? This sentence is not clear. I suggest rewriting it.

Answer: This place has been modified as suggested.

- Line 264: This information is repeated in the following discussion section. I believe the comparison between the results presented in the present paper and previously published models should be left to the discussion section.

Answer: The repeated sentence here has been deleted as suggested.

- Figures 8 and 9: Why there are blank spaces between layers? Is it a choice for displaying the model? If so, why is it presented like that? What were the criteria to display intervals with filled colors or as blank intervals? Why the blank intervals are sometimes thick in relation to the adjacent colored intervals and sometimes this relation is inverted?

Answer: The blank spaces are the dividing lines of the same time intervals. The density of the spaces here are mainly used to imply the sedimentation rate. When the blank space is thick, it generally means the sedimentation rate is relatively high. When the blank space is narrow, it means the sedimentation rate is relatively low.

---

## Author Response (AR3)

**Author's response to the comments**

1. The changes you make in response to the comment around efficiency ratio of deposition to erosion are useful, but to my mind you do not yet answer the part of the question about what the phrase "efficiency ratio of deposition to erosion" means. Please could you define what you mean by this, i.e. what efficiency means in this context?

Answer:Under a normal diffusion circumstance, if the transport coefficient is fixed, the changing rate of $h$ (the topography) is only related to the gradient of the current $h$, no matter if it is a depositional process or an erosional process. The "efficiency ratio of deposition to erosion" here is the efficiency ratio imposed besides the original diffusion process. With this parameter, we could enhance or weaken a certain process according to our need. In order to make it more clear, this place has been added in the text.

2. Your response to the reviewer regarding the comment beginning "What the authors mean by "hydraulic characteristic energy"…" is useful. Please could you integrate this explanation in to the manuscript.

Answer:This explanation has been added as suggested.

3. Your response to the author's comment beginning "I suppose there are things that Sedapp v2021 do better than the Sedpak model…" is again very useful, but not reflected in the manuscript. As a model development paper it is valuable to contrast with alternative tools. Please can you incorporate this response in to the paper, making sure you evidence the points being made.

Answer:This place has been incorporated in the text as suggested.

4. Your response to the reviewer comment beginning "Figures 8 and 9: Why there are blank spaces between layers…" is again something that would be useful to the reader. Could this be included in the figure caption?

Answer:This place has been included in the figure captions as suggested.

---

## Author Response (AR4)

**Topical Editor Decision: Publish subject to technical corrections (12 Jul 2021) by Paul Halloran**

**Comments to the Author:**

Dear Shuyu Sun and co-authors,

Thank you for your minor revisions. I would like to accept the manuscript after one final extremely minor technical revision.

The two new sentences "This seems to have little thing to do with the transportation capability originated from the river injection. It's more like the energy inherent in the basin that carries the sediments to the basin center." Read a little awkwardly, and are not completely clear to me.

Can I suggest that you modify the 1st sentence to something like:

"This seems to have little to do with the transportation capability originated from the river injection."

Then the 2nd sentence ("It's more like the energy inherent in the basin that carries the sediments to the basin center.") is not completely clear to me, and I think it is saying almost the same thing as the final sentence in the paragraph. If I have understood what you are trying to say, perhaps the end of the paragraph could read:

This seems to have little to do with the transportation capability originated from the river injection, instead the most appropriate description is as the energy inherent in the basin that carries the sediments to the basin centre.

Many thanks for making these final changes. Kind regards,

Paul Halloran

**Author's response:**

Thank you very much. These sentences have been modified as suggested.